



# Community-Scale Assessment of Flood-Related Public Health Vulnerability Using Multi-Criteria AHP in Northwestern Bangladesh

Nafisa Nuari Islam[1], Tonoy Mahmud[2], Shamima Ferdousi Sifa[2], Md. Asif Rafsan[2], A. S. M. Maksud Kamal[2], Md. Shakhawat Hossain[2], and Md. Zillur Rahman[2]

[1]Climate and Disaster Management, Jashore University of Science and Technology, Jashore-7408, Bangladesh
[2]Disaster Science and Climate Resilience, University of Dhaka, Dhaka-1000, Bangladesh

*Correspondence to:* Nafisa Nuari Islam, (nn.islam@just.edu.bd)

**Abstract.** Bangladesh faces heightened flood vulnerability due to climate change, particularly in riverine areas where health impacts are severe. The study aims to estimate the public health vulnerability in Dimla Upazila (Nilphamari district) using the Analytical Hierarchy Process (AHP), incorporating expert-weighted indicators: socio-demographics, WASH infrastructure, healthcare access, flood intensity, relief availability, and adaptation capacity. 315 households from six unions were randomly selected and structured survey were performed in 2019. Results reveal the contribution of all indicators to public health vulnerability whereas flooding intensity has been the most influencing factor, followed by relief accessibility, healthcare service accessibility, WASH infrastructure availability, and the adoption of adaptation capacity. The findings indicate that the northern region (Purba Chhatnai) exhibits the highest vulnerability due to its low socioeconomic status and limited access to relief services. In contrast, Tepa Kharibari—a centrally located union adjacent to the river—experiences frequent flooding but demonstrates moderate vulnerability owing to robust DRR measures, underscoring that physical exposure alone does not determine health risks. This study pioneers an integrated approach that connects household-level vulnerabilities, infrastructure robustness, and post-disaster responses to reveal previously unrecognized patterns in public health vulnerability in flood-prone regions worldwide. The findings demonstrate that combining proactive risk reduction with reactive emergency measures significantly enhances community health resilience, offering a transformative approach to disaster management. Furthermore, the research outcomes can assist policymakers in identifying gaps within the existing public health infrastructure of flood-prone areas, enabling the adoption of targeted DRR interventions to foster healthier and more resilient communities.

**Keywords:** public health, analytical hierarchical process (AHP), expert opinion, flood vulnerability, north-western Bangladesh





## 1 Introduction

Climate change has significantly amplified both the frequency and severity of flooding events, leading to escalating public health crises through the proliferation of waterborne and vector-borne diseases (Achanta et al., 2001), (Alam et al., 2018),(Ramin & McMichael, 2009). The health impacts of flooding extend far beyond immediate mortality, encompassing complex secondary effects including mass displacement, food insecurity, respiratory illnesses, and the destruction of critical

health infrastructure (Kovats et al., 2003),(WMO, 2015). These challenges are further compounded by socio-economic factors that disproportionately affect vulnerable populations, creating intricate patterns of health vulnerability across different communities (Ahern et al., 2005),(Du et al., 2010),(Phung et al., 2016). Bangladesh, which has a unique geographic position as a low-lying deltaic plain, combined with its extreme population density, makes it particularly vulnerable to recurrent flood disasters and its impacts (Afjal Hossain et al., 2011),(Rahman & Salehin, 2013). Approximately 35% of the whole country got

flooded in 2020 (NAWG Bangladesh, 2020) and the Sylhet flood of 2022 is termed as the worst flood in this region in 122 years (Rumpa et al., 2023).

The relationship between flood exposure and health outcomes has been demonstrated in various global contexts (Flores et al., 2024),(KaTurner et al., 2019),(Suhr & Steinert, 2022). For instance, as demonstrated in (Saulnier et al., 2018), a direct link

between seasonal flooding and health risks, but the key influencing indicators remain understudied, limiting the clarity needed for future interventions. A critical review (Lowe et al., 2013) of numerous PubMed studies was conducted and found that while factors like age, gender, socioeconomic status, and flood severity influence flood-related health outcomes, research remains limited on non-demographic risk factors. A research in Mombasa, Kenya revealed how household characteristics, WASH infrastructure, and environmental factors collectively influence health vulnerability during floods (Okaka & Odhiambo,2019a).

Similar dynamics are evident in Bangladesh, where studies have documented the catastrophic consequences of compromised WASH facilities during floods, including the widespread contamination of water sources and subsequent disease outbreaks (Shahid, 2010),(Shimi et al., 2010). Historical evidence from the 1998 floods showed that economically disadvantaged populations suffered disproportionately from diarrheal diseases due to limited access to clean water (Kunii et al., 2002).

While existing research has made valuable contributions to understanding flood-related health vulnerabilities, significant knowledge gaps remain. Previous studies have either focused on specific diseases (Hashizume et al., 2008),(Nahar et al., 2014),(Schwartz et al., 2006) or assessed e change impact on public health (Kundzewicz & Takeuchi, 1999),(Lee et al., 2021), (Shahid, 2010) or employed aggregated district-level analyses (Lee et al., 2021), potentially obscuring important variations at the household level. The 2021 study, while identifying regional vulnerability patterns in north-western Bangladesh,

acknowledged the limitations of district-level data in capturing micro-scale disparities. This highlights the critical need for research that examines vulnerability through a household-level lens, accounting for the complex interplay of structural, economic, and environmental factors that determine health outcomes during floods.

This study addresses these research gaps through three primary objectives: (1) developing a comprehensive household-level vulnerability assessment framework that integrates hazard exposure, demographic characteristics, WASH infrastructure, healthcare access, and adaptive capacity; (2) identifying spatial patterns of health vulnerability within flood-prone communities; and (3) evaluating the effectiveness of existing disaster risk reduction strategies. Our research makes several novel contributions to the field, including the development of a micro-scale assessment methodology that reveals intra-community vulnerability variations, and the demonstration of how combined proactive and reactive measures can significantly enhance health resilience. Focusing on the devastating 2020 floods that affected 35% of the country (NAWG Bangladesh, 2020), this study provides policymakers with actionable insights for targeted interventions, while offering a transferable framework for flood-vulnerable regions worldwide.

The findings of this research will enable government agencies, NGOs, and health professionals to identify critical gaps in current disaster response systems and develop more effective, localized strategies for reducing public health vulnerabilities during flood events. By bridging the divide between macro-level vulnerability assessments and household-level realities, this study aims to contribute to more equitable and effective disaster risk reduction policies in Bangladesh and similar flood-prone regions.

## 2 Material and Methods

This study employs a mixed-methods approach combining household surveys, expert judgments, and geospatial analysis to assess public health vulnerability in Bangladesh's Teesta River floodplain. Data were collected from 315 households across six unions in Dimla Upazila through structured questionnaires, capturing socio-demographic, WASH, healthcare access, flood conditions, and disaster adaptation indicators. The AHP was used to weigh vulnerability factors based on expert opinions, followed by normalization and aggregation to compute a composite vulnerability index. Statistical analyses (correlation, regression) identified key determinants of vulnerability, while GIS mapping visualized spatial patterns using Inverse Distance Weighting (IDW). The methodology integrates quantitative and qualitative insights to evaluate flood-induced health risks and inform resilience-building strategies.

### 2.1 Study Area

A significant number of rivers flow through the northern region of Bangladesh. The Teesta floodplain is one of the largest basins in the north-western part of the country (Mondal et al., 2021a). This region is highly vulnerable to floods occurring almost yearly due to its geographical location. The Nilphamari district is located upstream of the Teesta Basin, with significant rivers named Teesta, Dharla, and Brahmaputra flowing on the right side of the basin (Mondal et al., 2020). The right banks of the rivers are highly vulnerable and regularly affected by flood events, river bank erosion, and drought (BBS, 2020). Floods





in 1998, 2004, 2008, 2017, and 2020 are some of the significant and severe flood events that have devastated this area (Ahmed
& Hussain, 2009).

This study has focused on a small area of the Teesta floodplain, i.e., the Dimla Upazila (sub-district) (Fig01) which is located
in the north-eastern part of the Nilphamari District (Mondal et al., 2021b), with the Teesta River flowing through it (BBS,
2020). This Upazila is bordered by West Bengal, India on the north, Jaldhaka Upazila (under Nilphamari district) on the south,
Hatibandha Upazila (under Lalmanirhat district) on the east, and Domar Upazila (under Nilphamari district) on the west (BBS,
2013). The Gozaldoba Teesta Barrage in West Bengal, India, established for irrigation purposes, is on the Northern side of the
Dimla Upazila (Al-Hussain et al., 2021). The key informants during the field visit informed that when the rainfall increases in
the adjacent part of India (of the study area), the probability of flooding rises there and therefore they open the sluice gates of
the Gajoldoba Barriage to reduce their risk which results that the water flowing down to the northern part of Bangladesh
through the Brahmaputra River and cause flooding along the riverbank areas. This condition worsens if the rainfall in
Bangladesh continues as it increases the peak of the water level height. The Dimla Upazila is one of the worst affected regions
considering the number of affected people in frequent disasters (Rahman et al., 2018). It has an area of 326.80 sq. km. and the
male-female ratio of the area is 1.009:1 among the 283, 438 population (BBS, 2014). This Upazila consists of 10 union
parishads. Agriculture is the major livelihood option of the people of this area (BBS, 2014).



**Fig01: Study Area location (a) the Dimla Upazila location in the Northwestern part of Bangladesh along the Teesta floodplain; and (b) the surveyed households in different unions of the Dimla Upazila (illustrated by the author; field survey 2019)**

## 2.2 Sample size selection

A total of 315 samples were collected from six unions named Jhunagachh Chapani, Khalisha Chapani, Khoga Kharibari, Paschim Chhatnai, Purba Chhatnai, Gayabari and Tepakhribari of the Dimla Upazila (BBS, 2014). The Upazilas were selected purposely based on: low, moderate and high flood- affected regions according to the key informants (Upazila Nirbahi Officer (UNO—the executive head of the Sub District administration) and local NGOs (BRAC, ASA, and RDRS). In this study, to estimate the sample size, firstly the population has been projected for 2019 (the study year). The following formula (Goodman, 1968) has been used to project the population.

$$x(t) = x_0 \times (1 + r)^t$$

Here, $x(t)$ denotes the projected population. The data for $x_0$ is the population according to (BBS, 2020); r is the population growth rate (1% as (Chowdhury & Hossain, 2019)) and t is the time period (8 years). Since we have selected one surveyor





from each household to understand the picture of the whole, we have estimated the household number (for $x_0$) rather than the total population. The projected household number for 2019 is 67444.

After calculating the household number, the following equation (Smith, 2000) has been used to calculate the sample number.

*Necessary sample size= ((z-score)² x standard deviation x (1-Standard deviation)) / (margin of error)²*

For this research, we have taken a 90% confidence interval with a margin error of +/- 10%. Then the ideal sample size is 270. Since we cover six unions of the Dimla Upazila, we expanded the sample size and conducted a total number of 315 sample surveys. The survey numbers from each union have been tried to represent the population size of the unions correspondingly

(Table 1).

Table 1: Survey numbers according to union (representation of the presentation)

| Union | Population (According to the 2011 BBS Census) | Population Percentage | Sample Number | Sample Percentage |
|---|---|---|---|---|
| Gayabari | 5227 | 11.62 | 45 | 14.29 |
| Jhunagach Chapani | 11199 | 24.89 | 63 | 20 |
| Khalisha Chapani | 10446 | 23.22 | 75 | 23.81 |
| Khoga kharaibari | 5810 | 12.91 | 32 | 10.16 |
| Purba Chatnai | 4472 | 9.94 | 35 | 11.11 |
| Tepa kharibari | 7836 | 17.42 | 65 | 20.63 |
| **Total** | **44990** | **100** | **315** | **100** |

(Source: Illustrated by the author; field survey 2019)

**2.3 Data collection**

A field visit was conducted in October 2019 to collect household-level data through a structured questionnaire survey. Information related to socio-economic demography, WASH infrastructure availability healthcare service accessibility, flood hazard intensity, relief accessibility and adaptation strategies (Fig02) has been collected from 315 households followed by simple random sampling in six Upazilas. The questions were asked in the native language (Bangla) and each questionnaire took around 30 minutes. Each data variables were collected through binary responses (yes=1, no=0) and Likert scales (table-

2). Additional open-ended questions were also pursued to gain deeper insights into their perspectives and the broader circumstances of vulnerability. Participants were selected from each household, with preference given to the household head; alternatively, any adult family member aged 18 or older was included. The UNO, local NGOs (BRAC, ASA, RDRS) and Upazila Health & Family Planning Officer of the area were the key informants who provided the overview of the area in the initial phase of the survey.





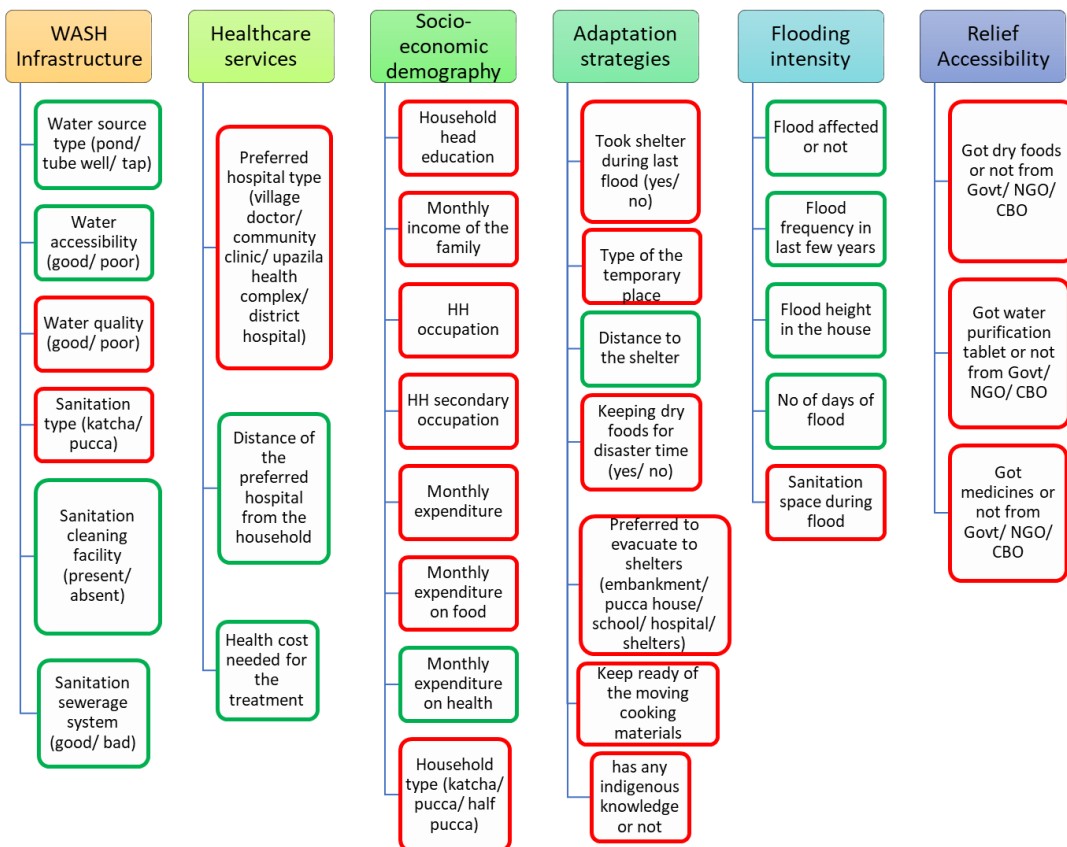

**Fig02: The indicators under the major categories taken for the vulnerability assessment (the green one shows the benefit criteria and the red one shows the cost criteria) (illustrated by the author; field survey 2019)**

There are six categories and 32 indicators in total to assess the vulnerability. Primarily, the categories were selected after reviewing the literature (Ahern & Kovats, 2006), (Acharya & Silori, 2024), (Few & Matthies, 2006), (Islam, 2017), (Laura

Tascón-González et al., 2020), (Matsuyama et al., 2020), (Okaka & Odhiambo, 2019a), (Okaka & Odhiambo, 2019b), (Paterson et al., 2018), (Phung et al., 2016), (Shah et al., 2020), (Shimi et al., 2010). After that, the experts help to finalize the indicators under the six categories. One indicator can influence the vulnerability in two ways: some accelerate the vulnerability and some lessen the vulnerability such as: better WASH facilities will lessen the public health vulnerability and vice versa (Fig02).

Table 2: The description of the variables of this study

| Variables | Descriptors | Measurement | Values |
|---|---|---|---|
| **Demographic characteristics** | | | |
| **Age** | Age of the respondent? | 18-30 | 20.30% |
| | | 31-40 | 28.50% |





| | | 41-50 | 23.50% |
|---|---|---|---|
| | | 51-60 | 14.40% |
| | | 60+ | 13.30% |
| **Gender** | Gender of the respondent? | Male | 64% |
| | | Female | 36% |
| **Marital Status** | Marital status of the respondent? | Married | 95.30% |
| | | Unmarried | 2.20% |
| | | Widow | 2.20% |
| | | Separated | 0.30% |
| **Education** | Education class of the respondent? | No Answer | 1.70% |
| | | Illiterate | 17.30% |
| | | Can sign, read, and write only | 24.30% |
| | | Class 2-9 | 32.40% |
| | | Secondary School Certificate (SSC) | 14.50% |
| | | Higher Secondary School Certificate (HSC) | 4.80% |
| | | Graduation | 5% |
| **Household Head's (HH) highest education level** | Highest education class of the household head? | Illiterate | 14% |
| | | can only sign | 20.70% |
| | | have primary education | 27.90% |
| | | 'Didn't complete secondary school (SSC) | 18.50% |
| | | completed secondary school (SSC) | 9.70% |
| | | completed higher secondary school (HSC) | 5.20% |
| | | Graduation | 4% |
| **HH primary occupation** | | Agriculture | 43.80% |



| | | | |
|---|---|---|---|
| | Primary Occupation of the household head? | Day labor | 22.6% |
| | | Business (both formal and informal) | 19% |
| | | Homemaker | 11.6% |
| | | Unemployed | 3% |
| **HH secondary occupation** | Secondary Occupation of the household head? | No secondary job | 65.40% |
| | | Day laborer/ Farming/ Fishing/ Poultry | 9.80% |
| | | Businessman | 18.70% |
| | | Informal economy | 4.80% |
| | | Formal economy | 1.30% |
| **Monthly income (BDT)** | Monthly income of the family? | 1500-5000 | 10.50% |
| | | 5001-10000 | 45.20% |
| | | 10001-20000 | 31.50% |
| | | 20001-35000 | 9.40% |
| | | 35000+ | 3.40% |
| **Monthly expenditure (BDT)** | Monthly expenditure of the family? | 2000-5000 | 13.40% |
| | | 5001-10000 | 55% |
| | | 10001-25000 | 29.80% |
| | | 25001-50000 | 1.50% |
| | | 50000+ | 0.30% |
| **Monthly expenditure on food (BDT)** | Monthly expenditure of the family on food? | 300-1500 | 2.60% |
| | | 1501-3000 | 21.80% |
| | | 3001-5000 | 36.60% |
| | | 5001-10000 | 33.10% |
| | | 10000-15000 | 5.30% |
| | | 15000+ | 0.60% |
| **Monthly expenditure on health (BDT)** | Monthly expenditure of the family on health issues? | 100-500 | 23.70% |
| | | 501-1500 | 46.70% |
| | | 1501-3000 | 22.80% |
| | | 3001-6000 | 5.90% |
| | | 6000+ | 0.90% |
| **Household type** | What is your household type | Thatched | 70.80% |



| | | Pucca | 5.10% |
|---|---|---|---|
| | | Semi-pucca | 24.10% |
| **Disease Outbreak** | | | |
| **Diarrhea** | Did your family members suffer from diarrhea during/ after the last flood? | Yes | 37.79% |
| | | No | 62.21% |
| **Fever** | Did your family members suffer from fever during/ after the last flood? | Yes | 8.12% |
| | | No | 91.88% |
| **Malaria** | Did your family members suffer from malaria during/ after the last flood? | Yes | 5% |
| | | No | 95% |
| **Typhoid** | Did your family members suffer from typhoid during/ after the last flood? | Yes | 2% |
| | | No | 98% |
| **Skin disease** | Did your family members suffer from skin disease during/ after the last flood? | Yes | 11.66% |
| | | No | 88.34% |
| **WASH infrastructure availability** | | | |
| **Water accessibility** | From where do you get water for maximum time of the year? | Tube well | 94.90% |
| | | Tap | 2.50% |
| | | Pond | 2.60% |
| **Water safety** | What is the quality of the water that you use? | Safe water | 76.20% |
| | | Unsafe water | 23.80% |
| **Water availability/ accessibility** | Do you get enough water for your daily purpose? | Good | 87.60% |
| | | Poor | 12.40% |
| **Sanitation type** | What is your sanitation type? | Thatched | 92.70% |
| | | Pucca | 7.30% |
| **Sanitation cleaning facilities** | How is your cleaning facility available? | Present | 37.50% |
| | | Absent | 62.50% |
| **Sanitation sewerage system** | How is your sanitation sewerage system? | Good | 43.50% |
| | | Poor | 56.50% |
| **Flooding severity** | | | |
| **Flood affected (in the last flood) or not** | Have you been affected by a flood in the last 5 years? | Yes | 55.60% |
| | | No | 44.40% |
| **Flood frequency in the last few years** | Can you tell me how many times a flood hit your house in the last 5 years? | 0 | 9.50% |
| | | 3-Jan | 81% |
| | | 6-Apr | 7.60% |
| | | 8-Jul | 1.90% |
| **Flood height in the house** | Can you tell me about the flood height in your house? | 0-2 | 55.20% |
| | | 5-Mar | 19.30% |



| | | 8-Jun | 3.40% |
|---|---|---|---|
| **Number of days of the flood** | How many days (max) you live in flood water in the last 5 years? | 0-5 | 73% |
| | | 10-Jun | 15.60% |
| | | 20-Nov | 5.70% |
| | | 21-45 | 3.50% |
| | | 45+ | 2.20% |
| **Sanitation option during floods** | What was your sanitation option during flood? | - | - |
| **Healthcare service accessibility** | | | |
| **Preferred hospital type** | In which hospital do you prefer to go for treatment? | Town hospital | 18.40% |
| | | Upazila Health complex | 14.90% |
| | | Community clinic | 24.40% |
| | | Village doctor | 41.90% |
| **Distance from household (KM)** | How far the preferred hospital is from your home? | 0-5 | 71.30% |
| | | 15-Jun | 13.50% |
| | | 16-25 | 6.30% |
| | | 26-50 | 2.50% |
| | | 50+ | 6.40% |
| **Treatment cost (BDT)** | How must cost is required for the treatment on an average each time? | 0-300 | 59.60% |
| | | 301-1000 | 18.80% |
| | | 1001-5000 | 17.90% |
| | | 5001-10000 | 2.80% |
| | | 10000+ | 0.90% |
| **Adaptation measures** | | | |
| **Took shelter during the last flood** | Did you take any shelter during the last flood? | Yes | 67.90% |
| | | No | 32.10% |
| **Type of the temporary place** | Where did you take place temporarily during a flood? | No | 32.10% |
| | | Embankment | 40% |
| | | 'Neighbor's house | 11.10% |
| | | 'Relative's house | 9.50% |
| | | Town | 3.80% |
| | | Others (school/ shelter) | 3.50% |
| **Distance to the shelter (KM)** | How far is the place from your home? | 0-2 | 93.70% |
| | | 7-Feb | 5.70% |





| | | 10-Aug | 0.30% |
|---|---|---|---|
| | | 10+ | 0.30% |
| **Preferred to evacuate to shelters (embankment/ pucca house/ school/ hospital/ shelters)** | Did you prefer to go to any shelter (school/ flood shelter/ pucca house/ embankment/ hospital etc.)? | Yes | 62.20% |
| | | No | 37.80% |
| **Keep dry foods for flood time** | Did you store dry foods for flood time? | Yes | 36.25% |
| | | No | 63.75% |
| **Keep ready the moving cooking materials** | Did you keep cooking materials with you during the evacuation? | Yes | 26.30% |
| | | No | 73.70% |
| **has any indigenous knowledge or not** | Do you have any sort of indigenous knowledge to flood? | Yes | 82.50% |
| | | No | 17.50% |
| **Relief Accessibility** | | | |
| **Got dry foods or not from Govt/ NGO/ CBO** | Did you get dry foods from GO/ NGO/ CBO/ or others? | Never | 69.80% |
| | | Sometimes | 19% |
| | | Always | 11.10% |
| **Got medicine or not from Govt/ NGO/ CBO** | Did you get medicines from GO/ NGO/ CBO/ or others? | Never | 89.20% |
| | | Sometimes | 8.90% |
| | | Always | 1.90% |
| **Got water purification tablet or not from Govt/ NGO/ CBO** | Did you get a water purification tablet from GO/ NGO/ CBO/ or others? | Never | 87.60% |
| | | Sometimes | 9.20% |
| | | Always | 3.20% |

(Source: Illustrated by the author; field survey 2019)

**2.4 Normalization of all indicators**

In the pre-processing stage, raw indicator values for each household were normalized to a common scale to facilitate
aggregation in the vulnerability assessment following multi-criteria analysis, as the factors are both quantitative and qualitative
with different units, they need to be harmonized for better comparison. Therefore, the linear max-min dimensionless method
for cost and benefit criteria has been used in this study as it has the advantage of removing the convertible units (Jahan &
Edwards, 2015).

Benefit criteria: $\quad n_{ij} = \dfrac{r_{ij} - r_j^{min}}{r_j^{max} - r_j^{min}}$

Cost criteria: $n_{ij} = \dfrac{r_j^{max} - r_{ij}}{r_j^{max} - r_j^{min}}$





The factors have been divided into two criteria based on their characteristics (Fig.02). Those that are positively impacted by the vulnerability are considered as the benefit criteria i.e. poor water quality increases the vulnerability of the household to different health hazards. Similarly, the factors that are inversely related to public health vulnerability are considered as cost criteria i.e. the presence of proper sanitation facilities decreases the vulnerability related to public health.

## 2.5 Analytical Hierarchy Process (AHP) method

The AHP method is one of the most significant and widely used weighting methods in the multi-criteria decision-making process by decision-makers and researchers (Kara, 2019), (Shim, 1989). It follows a pairwise comparison of all the considered categories to measure relatively, considering the value within 1-9 (Saaty scale) in the analysis (Saaty, 2014) where higher
values indicate stronger preference for Parameter A over B (Table 3). The eigenvector method was applied to convert pairwise comparisons (Table 3) into normalized criterion weights: for each expert, a square matrix A was created, where elements $a_{ij}$ represented the Saaty-scale preference of parameter *i* over *j*. For each expert, a square matrix AA was created, where elements $a_{ij}$ represent Saaty-scale preferences of parameter i over j.

Table 3: Expert-based pairwise comparison matrix of AHP parameters with Saaty-scale preferences

| No. | Parameter X | Parameter Y | Preferable Parameter (X/Y) | Degree of Preferences (Scale 1-9) |
|---|---|---|---|---|
| 1. | WASH facilities | Hospital facilities | - | [Value, e.g., 3] |
| 2. | | Household characteristics | - | - |
| 3. | | Adaptation strategies | - | - |
| 4. | | During disaster situation | - | - |
| 5. | | Access to relief | - | - |
| 6. | Hospital facilities | Household characteristics | - | - |
| 7. | | Adaptation strategies | - | - |
| 8. | | During disaster situation | - | - |
| 9. | | Access to relief | - | - |
| 10. | Household characteristics | Adaptation strategies | - | - |
| 11. | | During disaster situation | - | - |
| 12. | | Access to relief | - | - |
| 13. | Adaptation strategies | During disaster situation | - | - |
| 14. | | Access to relief | - | - |
| 15. | During disaster situation | Access to relief | - | - |

A total number of fifteen (15) experts from different domains took part in this assessment. These experts are academicians and practitioners from disaster management, environmental science, health science, and the public health domain. These experts are chosen based on their educational background (relevant domain), work experience in a suitable background (minimum 3





years), and their interests in the topic. They gave expert judgments by ranking the pairs from 1 (equal importance) to 9 (extreme
importance) following the Saaty scale (Table 4) as it is the most used and follows pairwise comparison comparing each
criterion with another, individually (Afshari et al., 2010).

Table 4: Brief explanation of the Saaty scale (Saaty, 2014)

| Intensity of importance | Definition |
|---|---|
| 1 | Equal importance |
| 3 | Weak importance of one over the other |
| 5 | Essential or strong importance |
| 7 | Demonstrated importance |
| 9 | Absolute importance |
| 2, 4, 6, 8 | Intermediate values between the two adjacent judgments |

The consistency ratio (CR) of all the judgments is kept within 10% to indicate a reasonable level of consistency within the
pairwise comparison (Afshari et al., 2010),(M. A. Chowdhury et al., 2021). If any individual expert's perception doesn't follow
the consistency ratio, then the whole process was rechecked, and then again consider the consistency ratio within 10%. The
consistency ratio is defined as:

$$CR = \frac{CI}{CR}$$

Here, the consistency index (CI) provides a measure of departure from consistency. The consistency index is calculated as:

$$CI = \frac{\lambda - n}{n - 1}$$

Where,

$\lambda$ = the average value of the consistency vector,

$n$ = the number of criteria

**2.6 Estimation of public health-related vulnerability**

The vulnerability is estimated by multiplying the weighted average (derived from the weighted sum model that has been used
by summing up the total weighted value of each category and then dividing it by the total number of indicators under the
category) with the normalized value of each indicator (Laura Tascón-González et al., 2020). Finally, all the values of the
categories are summed up and divided by the total number of the categories (six categories) to find the vulnerability value for
each household. Later, the average vulnerability was calculated for each union. The vulnerability was measured through the
following questions:



$$V_S = \sum_{i=1}^{n} \frac{(S_{1wn} + S_{2wn} + \dots\dots\dots + S_{kwn}) * S_{nv}}{k}$$

$$V_T = V_S + V_T + \dots + V_Z$$

Where,

S, T, ………., Z = Different categories

$S_{1,2,\dots,n}$ = Indicators under S category

$S_w$ = Weighted average of each indicator

$S_{nv}$ = Normalized value of S category

k = Total number of households

$V_T$ = Total Vulnerability

**2.7 Statistical Analysis**

In this study, several statistical analysis such as correlation, regression, and frequency analysis was adopted. Frequency
analysis has been performed to understand the overall scenario of the factors within the respondent groups. The regression
analysis is performed in two types: one is between the individual union vulnerability value with their category value and
another is between the public health vulnerability value and the categories (linear regression analysis).

$$y = \beta_0 + \beta_1 X + €$$

The linear regression is analysed with the above equation where $y$ is termed as the dependent variable (the target variable on
which the study is considering to imply) and $X$ is termed as the independent or explanatory variable (the input datasets which
are self-sustaining). $\beta_0$ and $\beta_1$ are the parameters of the model (these are the intercepts and reflect the relationship between
the dependent and independent variables). The parameter $\beta_0$ is termed as an intercept and $\beta_1$ is termed as the slope parameter.
The unobservable error component $€$ accounts for the failure of data (error) to lie on a straight line and represents the
difference between the true and observed realization of y. In the analysis of public health vulnerability and the categories, the
dependent variable is the vulnerability value and the independent variable is the six category values. In the other analyses (the
regression has been run separately), the individual categories are the dependent variables and the individual union vulnerability
value is the independent variable. In the linear regression analysis between the categories and the union-wise public health
vulnerability value, the average value of the categories for the individual unions has been taken. Moreover, the correlation has
been performed among the six categories.

**2.8  GIS Analysis**

The study area map and the vulnerability maps have been prepared in ArcGIS 10.8 environment. The union vulnerability is
shown as the average value of the unions. In the case of the household vulnerability map, IDW tool has been used to make the
vulnerability value to raster data. The classes have been determined manually.



## 3 Results

### 3.1 Demographic characteristics

Among the studied households (Table 2), most of the household heads (64%) were male, and the remaining (36%) were female. The highest (28.5%) number of respondents were in the age group of 31–40 years, followed by 23.5% of 41–50 years and
245 14.4% of 51–60 years. The rest were below 31 age or above 60 ages. Among the respondents, 95.3% were married, 2.2% were found unmarried, 2.2% were widows and 0.3% were separated. In terms of principal livelihood, the highest (43.8%) were involved in agriculture (farming, fishing, and poultry), followed by 22.6% of day labourers, 19% were involved in business activities (both informal and formal altogether), 11.6% were housewife and only remaining 3% were unemployed. As regards the level of literacy, almost 28.2% of them attended/ completed secondary school education and 27.9% had primary education.
Also, 14% of the household heads were illiterate, around 20.7% could sign, read, and write only, 5.20% passed the higher school certificate and only 4% were graduated. The monthly average household income was found Tk. 13,803. The highest 45.2% of the households' income was BDT 5001-10000, followed by 31.5% of BDT 10001 to 20000, 10.5% BDT 1500-5000, 9.4% of 20001-35000, and the lowest only 3.4% had BDT 35000+.

### 3.2 Disease outbreak after a flood in the study area and the surrounding factors that affect it

Different types of water-borne diseases are very common in flood-prone areas after any flood event. Among them, diarrhoea, typhoid, skin diseases, dysentery, cholera, and meningitis are very common (Shahid, 2010). Along with these, cold symptoms along with fever are regular during the flood. The impacts are not even limited to these; death from drowning, flash flooding, injuries and wounds, electric shocks, burns and explosions, hypothermia, and psychological effects are also some of the impacts of floods (Laura Tascón-González et al., 2020). In the study area (Fig03), the respondents reported that diarrhoea (>35%) is
the most common phenomenon during and after flood occurrence. Skin diseases (>10%) and fever (~ 9%) were also found to be common in the area. Around 5% and 1% of respondents reported that they suffered from malaria and typhoid during the last flood, respectively. They visit doctors or other health complexes depending on the severity of the health situation.

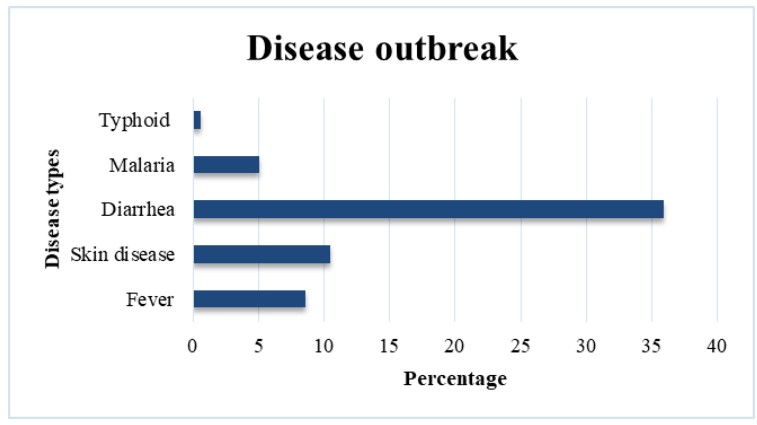

**Fig03: Percentage of diseases affected in the study area**

(Source: Illustrated by the author; field survey 2019)



Table 5 summarizes the prevalence (%) of five diseases—diarrhea, malaria, fever, typhoid, and skin disease—across different age groups and genders. Diarrhea was the most common ailment, particularly among individuals aged 20–35 (88.32%) and males exhibited a higher prevalence of diarrhea (87% vs. 76.5%). Malaria showed higher prevalence in the 20–35 age group (16.21%) and was more frequent in males (13.68%) than females (2.94%). Fever was substantially greater in females (32.35%) compared to males (14.54%). Typhoid had low overall prevalence, with the highest rate in the 46–60 age group (2.83%) and was rare in males (0.8%) but slightly more prevalent in females (2.9%), while skin disease was most common in the 26–45 age group (33.33%) but equally prevalent across genders (20.5%). These findings suggest variations in disease susceptibility based on age and sex.

Table 5: Disease distribution among different demographic people (given in percentage)

|  | Diarrhea | Malaria | Fever | Typhoid | Skin Disease |
|---|---|---|---|---|---|
| **Age** | | | | | |
| 20-35 | 88.32 | 16.21 | 22 | 1.7 | 21 |
| 26-45 | 75.28 | 8.34 | 13.8 | 0 | 33.33 |
| 46-60 | 82.71 | 0 | 15.79 | 2.83 | 15.79 |
| 61-75 | 76.19 | 10.15 | 19.05 | 0 | 14.29 |
| **Gender** | | | | | |
| Male | 87 | 13.68 | 14.54 | 0.8 | 20.5 |
| Female | 76.5 | 2.94 | 32.35 | 2.9 | 20.5 |

(Source: Illustrated by the author; field survey 2019)

### 3.3 Union-wise public health vulnerability calculation

The union-level assessment of public health vulnerability yielded unexpected findings, as illustrated in Fig04. While existing flood vulnerability studies (Ferdous & Mallick, 2019) identify the Brahmaputra River's north-western bank in Dimla Upazila as particularly flood-prone, our analysis reveals a distinct pattern in health outcomes. Key informant interviews confirmed that Tepa Kharibari Union experiences the most severe flooding due to its geographical positioning along the river, with some residents living on the opposite bank. However, contrary to expectations, our results demonstrate that Purba Chhatnai - the northernmost studied union - exhibits the highest public health vulnerability. The second most vulnerable area was Jhunagach Chapani in the extreme south, while Gayabari, located centrally at a distance from the riverbank, showed the lowest vulnerability among the studied unions. This spatial distribution suggests that flood exposure alone does not fully determine health vulnerability, highlighting the importance of other contributing factors.





**Fig04: Public Health Vulnerability map according to the studied unions**

(Source: Illustrated by the author: field survey 2019)

The analysis reveals that Purba Chatnai's position as the most vulnerable union stems from multiple interrelated factors (Table 3). The area's pronounced socio-economic challenges - with approximately 20% of residents living in extreme poverty (Ferdous & Mallick, 2019) - directly contribute to inadequate household conditions and substandard WASH facilities. Existing literature and our findings both confirm that while various adaptation measures have been implemented across Dimla Upazila, Purba Chatnai's adaptation capacity remains particularly weak (Ferdous & Mallick, 2019). Compounding these issues, the union receives disproportionately less attention in relief distribution despite its flood exposure, collectively exacerbating its public health vulnerability.

Jhunagach Chapani's status as the second-most vulnerable union reflects its dual burden of severe flood impacts coupled with minimal adaptation capacity and restricted relief access. In contrast, Tepa Kharibari demonstrates how institutional support can mitigate vulnerability - despite being the most flood-affected union (Table 6), its moderate public health vulnerability





ranking stems from robust NGO interventions that enhance both adaptation strategies and relief accessibility. Gayabari presents an interesting case where relative economic advantage and social capital translate to both superior adaptation approaches and easier relief procurement, despite its lower baseline flood risk. Key informant interviews corroborate these spatial patterns,

confirming Tepa Kharibari's high flood exposure alongside Gayabari's comparatively protected position.

Table 6: Comparative vulnerability metrics across study unions

| Union Name | WASH infrastructure | Socio- economic demography | Healthcare services | Relief accessibility | Adaptation strategies | Flooding intensity |
|---|---|---|---|---|---|---|
| Gayabari | 6.14 | 11.91 | 10.09 | 17.31 | 6.06 | 2.94 |
| Jhunagach Chapani | 6.28 | 11.78 | 10.71 | 18.69 | 6.54 | 4.84 |
| Khalisha Chapani | 6.33 | 14.01 | 10.95 | 17.17 | 6.40 | 4.34 |
| Khoga kharaibari | 6.78 | 14.40 | 10.41 | 17.12 | 6.29 | 4.52 |
| Purba Chatnai | 6.34 | 15.29 | 10.12 | 18.00 | 6.78 | 4.44 |
| Tepa kharibari | 6.42 | 13.15 | 10.75 | 17.21 | 5.35 | 5.23 |

(Source: Illustrated by the author; field survey 2019)

Table 6 provides empirical support for the observed vulnerability patterns, revealing significant variations across unions. Purba Chatnai emerges as the most vulnerable area, exhibiting the highest household deprivation score (15.29) and relatively poor relief accessibility (18.00), despite reporting higher adaptation strategy scores (6.78) - a paradox suggesting potential gaps between planned and implemented measures. Tepa Kharibari presents an interesting case of resilience, maintaining only moderate vulnerability despite recording the most severe flooding conditions (5.23), which appears mitigated by better hospital

facilities (10.75) and relief access (17.21). Notably, its surprisingly low adaptation strategy score (5.35) implies that institutional support may compensate for individual preparedness. Jhunagach Chapani's position as second-most vulnerable aligns with its combination of extreme flooding (4.84) and limited relief access (18.69), while Gayabari's advantage becomes clear through its minimal flood exposure (2.94) and superior relief accessibility (17.31). The data particularly highlights how relief systems (ranging 17.12-18.69) and hospital access (10.09-10.95) show less variation than household characteristics

(11.78-15.29) or flood severity (2.94-5.23), suggesting these structural factors may serve as critical equalizers in vulnerability



mitigation. These findings reinforce that while flood intensity establishes baseline risk, the interplay of socio-economic conditions and institutional support systems ultimately determines health outcomes.

However, the vulnerability map (Fig05) clearly demonstrates that riverside areas exhibit significantly higher vulnerability levels, particularly along the Brahmaputra's right bank where communities face very high to extreme public health risks. In contrast, the inland union surrounding Gayabari maintains consistently low vulnerability scores. Field interviews with right bank residents confirm this disparity, revealing critical service gaps that force dependence on left bank facilities during emergencies. Respondents reported lacking essential infrastructure on the right bank, necessitating hazardous cross-river movements during floods that compound their health vulnerabilities. This spatial analysis underscores how geographic isolation from core union services exacerbates vulnerability, even within the same administrative boundaries, highlighting the need for targeted interventions in these high-risk riverside settlements.

**Fig05: Public Health Vulnerability map of the studied unions (according to household vulnerability)**

(Source: Illustrated by the author: field survey 2019)



## 3.4 Relationship between the categories and public health vulnerability

In this study, the six categories can be divided into three disaster management phases: the pre-disaster or normal period, during disaster period, and the post-disaster recovery period (Fig06).

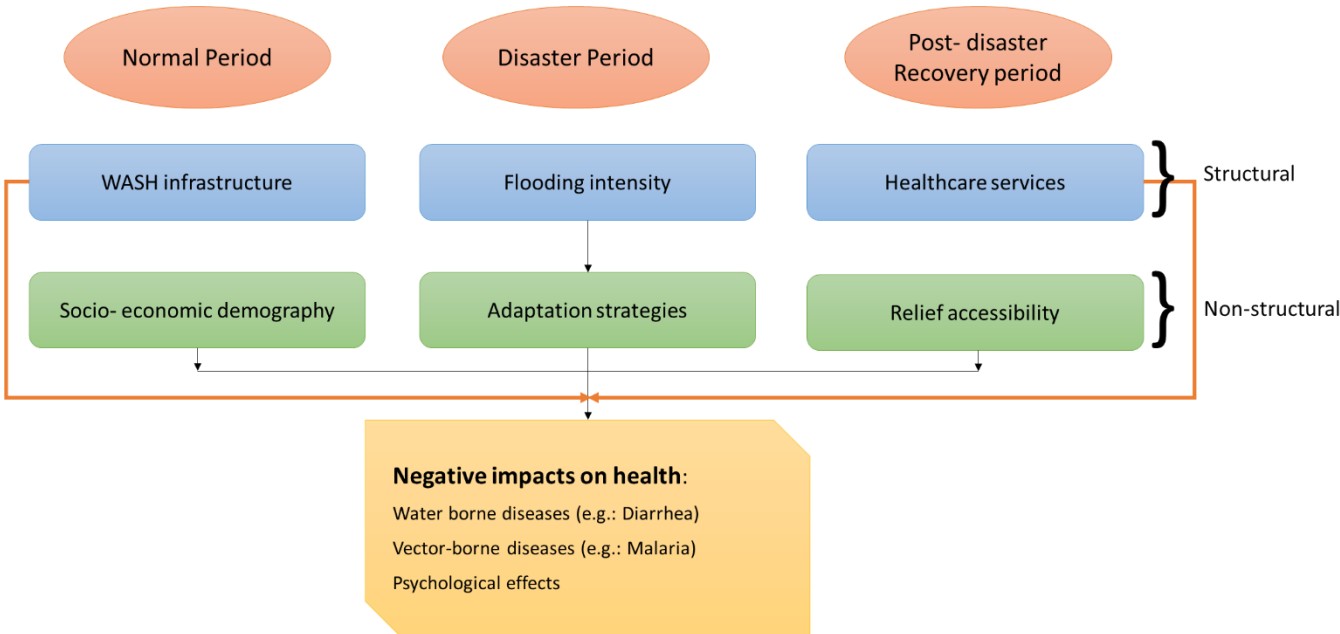

**Fig06: Flow chart showing the categories affecting the health status**

(Source: Illustrated by the author; field survey 2019)

Household socio-economic characteristics—including the education, occupation, income, and health expenditure of the household head—play a critical role in shaping health outcomes during both normal and pre-disaster periods. Additionally, access to proper Water, Sanitation, and Hygiene (WASH) infrastructure is essential for maintaining good health. During

disasters, the severity of flood conditions and the adoption of adaptation strategies (e.g., food storage) significantly influence household resilience and health. Post-disaster recovery, meanwhile, depends on healthcare accessibility and relief availability, collectively determining overall health quality.

These factors can be categorized into structural (WASH facilities, disaster-induced housing conditions, sanitation status, and healthcare infrastructure) and non-structural elements (household socio-economic status, adaptation strategies, and relief

access). Structural and non-structural components interact dynamically: improved household conditions facilitate better WASH infrastructure, enhancing disaster resilience, whereas economic constraints exacerbate vulnerability by limiting adaptive capacity and healthcare access.




The correlation analysis (table 7) reveals significant relationships among key resilience factors. A weak but positive correlation exists between relief accessibility and adaptation strategies (r = 0.098, p = 0.027), suggesting that better relief access may slightly improve adaptive capacity. More notably, adaptation strategies show a moderate negative correlation with flooding state (r = -0.254, p < 0.001), demonstrating their effectiveness in mitigating flood impacts. Household characteristics also negatively correlate with flooding state (r = -0.115, p = 0.042), indicating that socioeconomically advantaged households experience less severe flood effects. Most prominently, WASH facilities strongly associate with household characteristics (r = 0.031, p = 0.002), highlighting how better socioeconomic status enables improved WASH infrastructure, a crucial determinant of health resilience. These findings collectively emphasize the interconnectedness of structural and non-structural factors in shaping disaster resilience and health outcomes.

Table 7: Correlation between the categories

| Categories | Pearson Correlation | Significance |
|---|---|---|
| Relief Accessibility Adaptation strategies | 0.098* | 0.027 |
| Adaptation strategies Flooding State | -.254** | 0.000 |
| Household Characteristics Flooding State | -0.115* | 0.042 |
| WASH facilities HH characteristics | 0.031** | 0.002 |

*Correlation is significant at the 0.05 level (2-tailed)

**Correlation is significant at the 0.01 level (2-tailed)

(Source: Illustrated by the author; field survey 2019)

The relationship between public health vulnerability and the six categories has been examined through linear regression analysis (Table 8). The model shows that it includes 99.6% of data ($R^2$) and it demonstrates that all measured factors significantly contribute to public health vulnerability (p < 0.01 for all variables). Flooding intensity shows the strongest association (β = 0.037), indicating that worsening flood conditions increase health vulnerability most substantially. Relief accessibility (β = 0.034), healthcare services (β = 0.033), and WASH infrastructure (β = 0.032) exhibit nearly equal impacts, followed closely by socioeconomic factors (β = 0.031) and adaptation strategies (β = 0.029). The consistent positive coefficients reveal a dose-response relationship where each unit deprivation in any factor proportionally increases health vulnerability. These findings quantitatively confirm that both structural (WASH, healthcare) and non-structural (socioeconomics, adaptation) elements collectively determine community resilience, with flood intensity being the most





critical risk amplifier. The results underscore the need for integrated interventions addressing all these dimensions to effectively reduce public health vulnerability. However, all of these factors hold only a 1% error, which makes it clear that all these factors have a great influence on public health vulnerability.

Table 8: Relationship between the categories and public health vulnerability

| Categories | Measurement (B) | Measurement statement | Significance (p) |
|---|---|---|---|
| WASH infrastructure | .032 | One unit deprivation of WASH facilities will increase 0.032 units of public health vulnerability | 0.00 |
| Socio-economic demography | .031 | One unit decrease in the quality of HH characteristics will increase 0.031 units of public health vulnerability | 0.00 |
| Relief accessibility | .034 | One unit deprivation of access to relief will increase 0.034 unit of public health vulnerability | 0.00 |
| Adaptation strategies | .029 | One unit absence of adaptation strategies will increase 0.029 units of public health vulnerability | 0.00 |
| Flooding intensity | .037 | One unit worsening of disastrous (flood) condition will increase 0.037 units of public health vulnerability | 0.00 |
| Healthcare services | .033 | One unit deprivation in hospital facilities will increase 0.033 units of public health vulnerability | 0.00 |

(Source: Illustrated by the author; field survey 2019)

## 4 Discussion

This study reveals a critical insight: while flood intensity significantly impacts household health, other factors—such as relief accessibility, disaster preparedness, and institutional support—play an equally vital role in mitigating vulnerability. A striking
example from this study is Tepa Kharibari Union, which experiences the highest flood frequency and severity (Fig.03) yet



exhibits lower public health vulnerability due to robust government and NGO interventions, including timely relief distribution and community resilience programs. This underscores that both proactive (pre-disaster) and reactive (during/post-disaster) measures are indispensable for safeguarding health. Proactive initiatives (e.g., training, soft loans, awareness campaigns) enhance adaptive capacity, while reactive measures (e.g., emergency relief, medical aid) ensure immediate recovery—

demonstrating that an integrated approach maximizes resilience.

To systematically reduce flood-induced health risks, all six identified factors (WASH infrastructure, socioeconomic conditions, relief systems, adaptation strategies, flood severity, and healthcare access) must be embedded in national and local development plans. Bangladesh's Climate Change Strategy and Action Plan (CCSAP) already acknowledges climate-related health threats (Shahid, 2010), but further action is needed:

1. Enhanced WASH and Healthcare Systems – Ensure safe water, sanitation, and hygiene facilities alongside equitable access to medical services, particularly in flood-prone zones
2. Risk-Inclusive Urban Planning – Implement elevated housing (plinth levels), flood-resistant shelters, and hazard-zoned land use to minimize exposure

3. Multi-Stakeholder Collaboration – Strengthen partnerships between government agencies, NGOs, and communities to deliver:
   a. Pre-disaster: Training, early warning systems, and livelihood support (e.g., soft loans)
   b. During-disaster: Emergency WASH kits, mobile clinics, and evacuation protocols
   c. Post-disaster: Targeted relief distribution and mental health support

4. Community-Centric Approaches – Adopt bottom-up planning to align interventions with local needs, emphasizing food security, adaptive farming, and hygiene education

## 5 Conclusions

Bangladesh's pursuit of 'Vision 2041' hinges on fostering a healthy and resilient society through equitable access to healthcare

and robust WASH infrastructure. However, as a disaster-prone nation, Bangladesh faces significant public health risks exacerbated by floods and other hazards, necessitating a comprehensive assessment of vulnerability drivers. Achieving Vision 2041's goal of a healthy, resilient Bangladesh demands dual emphasis on prevention and response. By prioritizing inclusive policies, infrastructure hardening, and cross-sector coordination, the nation can transform vulnerability into resilience— ensuring that even the most flood-affected communities, like Tepa Kharibari, thrive despite climatic adversities. The lessons

here extend beyond Bangladesh, offering a blueprint for integrated disaster-health governance in vulnerable regions worldwide.

**Author Contribution**

Nafisa Nuari Islam: conceptualization, methodology, data curation, analysis, investigation, writing -original draft.



Tonoy Mahmud: methodology, data curation, investigation, writing- review, and editing.

Shamima Ferdousi Sifa: methodology, data curation, writing- review, and editing.

Md. Asif Rafsan: methodology, data curation, writing- review.

A. S. M. Maksud Kamal: supervision, resources, conceptualization.

Md. Shakhawat Hossain: supervision, resources, conceptualization.

Md. Zillur Rahman: supervision, resources, conceptualization, writing- editing.

## Competing interests

The author(s) declare no competing interests.

## Data availability

The research data can be shared upon reasonable request, subject to ethical approvals.

## Ethics approval

The author ensures that the work described has been carried out following The Code of Ethics of the World Medical

Association (Declaration of Helsinki) for experiments involving humans. (Ref: ERC/FBST/JUST/2023-138: Faculty of

Biological Science and Technology, Jashore University of Science and Technology).

## Acknowledgment

This research was conducted with the essential support of OXFAM Bangladesh, whose field coordination enabled systematic

data collection. We acknowledge with profound appreciation the cooperation of all survey participants and key stakeholders

in Dimla Upazila. The study's analytical rigor was made possible by fifteen dedicated experts who meticulously completed the

pairwise comparison matrices, and their contributions are gratefully recognized.

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
