# Peer review of "Community-Scale Assessment of Flood-Related Public Health Vulnerability Using Multi-Criteria AHP in Northwestern Bangladesh"

_EGUsphere, 2025_

## Referee Comment (RC1)

The manuscript entitled "Community-Scale Assessment of Flood-Related Public Health Vulnerability Using Multi-Criteria AHP in Northwestern Bangladesh" presents a significant and well-structured contribution to the field of natural hazards, particularly in the domain of hydrological disasters and their public health impacts. The study findings successfully capture real-life health challenges from a highly localized perspective, and the authors are to be commended for their thorough and insightful work. The questions from the review criteria were answered below:

1. The paper addresses scientifically and technically relevant questions within the scope of Natural Hazards and Earth System Sciences (NHESS), specifically focusing on floods—one of the major hydrological hazards. It also engages with thematic areas central to the journal, including risk assessment, mitigation and adaptation strategies, and socio-economic and management aspects of disasters.

2. The research offers novelty through an integrated approach linking household-level vulnerabilities, infrastructure robustness, and post-disaster responses. This enables the identification of previously unrecognized patterns in public health vulnerability in flood-prone regions, providing a methodological framework that is transferable to other areas with similar risk profiles.

3. The study adheres to international research standards.

4. All scientific methods and assumptions are valid and outlined clearly in the methodology section.

5. The use of statistical validation and mapping strengthens the reliability of the results, which adequately support the interpretations and conclusions.

6. The authors reach substantial conclusions with the findings.

7. Absolutely and very clearly, the authors have presented.

8. The title of the paper is self-explanatory, clear, and wonderfully captures the contents of the paper. Seeing the title, we get a brief idea about the study.

9. The abstract provides a concise and complete summary of the study's objectives, methodology, major findings, novelty, and applicability. However, the methodological component in the abstract could be made slightly more explicit for clarity.

10. The title and the abstract are pertinent and easy to understand to a wide and diversified audience, as the authors have used simple, clear, and relevant words.

11. The methodology is clearly explained. All the study instruments, methods, variables, and formulae are clearly explained and elaborated.

12. adequate.

13. Yes

14. appropriate. In-text references should be followed by the journal guidelines. Just have to put multiple references in one parenthesis using a semicolon, as stated in the guideline.

15. yes
16. yes. The authors have used a very simple way to present their study, which is very well appreciated.
17. adequate
18. The introduction is well-crafted, outlining the relevance and background of the study. Nevertheless, it would benefit from:
    - A brief discussion of current global and Bangladesh-specific perspectives on health vulnerability in flood-prone areas, to better contextualize the study's significance.
    - The inclusion of additional literature on health vulnerabilities in Bangladesh from a local perspective, to situate the research within existing scholarship.
    - Minor Issues:

    Formatting: In-text citations should follow the journal's style guide, with multiple references in a single parenthesis separated by semicolons.

    Font type should be kept the same throughout the paper.

    Typographical error: On page 24, line 390, the abbreviation "BCCSAP" is missing the initial "B."

19. Yes
20. Yes.

**Overall Assessment**

This is an excellent, well-organized study with substantial conclusions, clear presentation, and practical applicability.

The paper is clearly written, with a concise and informative title, a well-structured abstract (though the methodology could be stated more explicitly), and a logically organized methodology and results section. Statistical validation and mapping strengthen the conclusions, which are both substantial and well-supported. The integration of local-level data with methodological rigor makes it a valuable contribution to the literature on flood-related public health vulnerability.

Minor revisions are required, including:

1. Expanding the introduction to incorporate global and Bangladesh-specific perspectives on health vulnerability and relevant local literature.

2. Correcting minor typographical errors (e.g., "BCCSAP" on p.24, line 390).

3. Ensuring in-text citations conform to journal formatting guidelines (multiple references separated by semicolons).

**Recommendation:** With minor revisions as noted, the manuscript is suitable for publication.